# Virulence Factors of *Mycobacterium tuberculosis* as Modulators of Cell Death Mechanisms

**DOI:** 10.3390/pathogens12060839

**Published:** 2023-06-18

**Authors:** Lucero A. Ramon-Luing, Yadira Palacios, Andy Ruiz, Norma A. Téllez-Navarrete, Leslie Chavez-Galan

**Affiliations:** 1Laboratory of Integrative Immunology, Instituto Nacional de Enfermedades Respiratorias “Ismael Cosío Villegas”, Mexico City 14080, Mexico; ramonluing@yahoo.com.mx (L.A.R.-L.); doreydna@gmail.com (A.R.); 2Escuela Militar de Graduados de Sanidad, Secretaría de la Defensa Nacional, Mexico City 11200, Mexico; yadpal@gmail.com; 3Department of Biological Systems, Universidad Autónoma Metropolitana, Campus Xochimilco, Mexico City 04960, Mexico; 4Department of Healthcare Coordination, Instituto Nacional de Enfermedades Respiratorias “Ismael Cosío Villegas”, Mexico City 14080, Mexico; norma.tellez@gmail.com

**Keywords:** tuberculosis, cell death, virulence factors, apoptosis, necrosis, pyroptosis, necroptosis

## Abstract

*Mycobacterium tuberculosis* (Mtb) modulates diverse cell death pathways to escape the host immune responses and favor its dissemination, a complex process of interest in pathogenesis-related studies. The main virulence factors of Mtb that alter cell death pathways are classified according to their origin as either non-protein (for instance, lipomannan) or protein (such as the PE family and ESX secretion system). The 38 kDa lipoprotein, ESAT-6 (early antigen-secreted protein 6 kDa), and another secreted protein, tuberculosis necrotizing toxin (TNT), induces necroptosis, thereby allowing mycobacteria to survive inside the cell. The inhibition of pyroptosis by blocking inflammasome activation by Zmp1 and PknF is another pathway that aids the intracellular replication of Mtb. Autophagy inhibition is another mechanism that allows Mtb to escape the immune response. The enhanced intracellular survival (Eis) protein, other proteins, such as ESX-1, SecA2, SapM, PE6, and certain microRNAs, also facilitate Mtb host immune escape process. In summary, Mtb affects the microenvironment of cell death to avoid an effective immune response and facilitate its spread. A thorough study of these pathways would help identify therapeutic targets to prevent the survival of mycobacteria in the host.

## 1. Introduction

*Mycobacterium tuberculosis* (Mtb) is a member of the *Mycobacterium tuberculosis* complex (MtbC) and the primary causative agent of tuberculosis (TB) in humans. *Mycobacterium africanum* and *Mycobacterium bovis* also induce TB in humans under specific conditions, such as immunosuppression [1,2]. According to the World Health Organization (WHO), despite a reduction in the number of new TB cases in 2020 and 2021, the last WHO report (2022) showed an increase in TB-related deaths. This observation could possibly be attributed to the overestimation of the TB reduction, mainly due to undiagnosed TB due to disrupted access to TB diagnosis as a consequence of the COVID-19 pandemic [3]. 

Mtb is an intracellular pathogen, mainly infecting the respiratory tract; initially, it specifically infects resident alveolar macrophages (AMs) and replicates to survive intracellularly by avoiding the host immune response [4]. The innate immune cells in the lungs are the earliest defenders; they engulf the pathogen, degrade it, and regulate inflammation, whereas adaptive immune cells, such as CD4 and CD8 T cells, engage in the defense through an antigen-specific T cell response [5]. 

Cell death is an active process that critically regulates embryonic development, maintaining tissue homeostasis and eliminating damaged tissue and potentially harmful cells [6]. Although cell death is considered generally destructive, it is also a defensive process to preserve homeostasis. However, simultaneously activating several types of cell death could be detrimental to the organism that can be attributed to the induction of a hyper-inflammatory process [7,8]. 

Some Mtb antigens block apoptosis and activate other cell death pathways, such as necrosis, which allow its spread, thereby modulating cell death as a survival mechanism [9]. Currently, how Mtb leverages cell death to evade the host immune response is not entirely understood. However, a growing body of evidence indicates that virulence is a critical determining factor in the type of cell death triggered or inhibited by Mtb [10,11,12]. 

This review analyzes Mtb virulence factors that directly induce and modulate cell death pathways. This knowledge will be crucial for identifying critical cell death regulators that Mtb may inhibit to help develop therapeutic strategies. 

## 2. Mtb Virulence Factors: Who Are Involved in Disturbing Cell Death?

Virulence is the ability of a pathogen to cause disease [2]. To define Mtb virulence, the following factors should be considered: (1) the ability of a bacterium to avoid the host’s immune response; (2) its capacity to cause lung damage; and (3) its successful transmission to infect a new host [2,13]. 

Unlike other pathogens, such as *Vibrio cholerae* or *Corynebacterium diphtheriae*, Mtb does not use toxins and enzymes, which are the typical virulence factors; instead, several virulence-associated genes compensate for them [2].

Mycobacterial virulence factors can be divided based on their nature into (a) non-protein molecules, such as lipids, sugars, and (b) proteins. The following Mtb molecules can modify cell death pathways:

### 2.1. Non-Protein Virulence Factors

Lipids, glycolipids, glycans, nucleic acids, and metabolites are included in this group; many are vital cell surface components involved in host–pathogen interaction, recognition, intracellular survival, and virulence [14]. 

#### Major Non-Protein Virulence Factors Inside the Cell Wall

Mtb has a lipid-rich envelope essential for its survival and virulence [14]. The mycobacterial cell wall contains up to 60% lipids [15], and many cell wall components that are secreted, shed, or localized on the bacterial surface, interact with the host cells [16]. Before describing how virulence factors modulate the cell death pathway, a brief description of the more abundant non-protein ones is provided below:Phosphatidyl-myo-inositol mannosides (PIMs) are the most abundant glycolipids in the mycobacterial cell envelope and are precursors of lipomannan (LM) and lipoarabinomannan (LAM) [17]. The PIMs comprise variable numbers of mannose units and levels of acylation. Virulent species possess PIMs with five or six mannoses that bind to the mannose receptor (MR), contributing to macrophage uptake. A few of the mannoses also interact with the dendritic cell (DC)-specific intercellular adhesion molecule-3-grabbing non-integrin DC-SIGN from DC [2,18]. Acyl phosphatidyl-myo-inositol dimannoside (AcPIM2) is a part of the inner membrane in most mycobacterial species, and AcPIM6 is involved in maintaining cell envelope integrity [16];LM is a multi-glycosylated lipid or polymannosylated PIMs, which is the basic structure of LAM. LM efficiently activates the innate immune response via a tetra-acylated form that activates macrophages through toll-like receptor-2 and 4 (TLR2 and TLR4, respectively). In contrast, its di-acylated form regulates and inhibits nitric oxide (NO) production and cytokine secretion in activated macrophages [19];LAM is a glycolipoconjugate composed of LM bound to multiple arabinose residues. When LAM acquires extra and random formation of mannose-capped LM, it is called ManLAM. An addition of phosphoinositol-capped LM makes it known as PILAM, and uncapped or arabinofuranosyl-terminated LM makes it AraLAM. PIMs, LM, LAM, and ManLAM, are recognized by specific receptors, expressed on the cell surface of antigen-presenting cells, such as MR, DC-SIGN, and dendritic cell activating receptor (DCAR), facilitating Mtb uptake into host cells [20,21]. Indeed, the diversity in Mtb mannosylated cell walls alters its virulence, affecting pathogeny and host adaptation [21];Phthiocerol dimycocerosates (PDIM) are lipids present on the outer leaflet of the outer membrane that may be secreted or shed from mycobacteria. PDIM promotes Mtb’s evasion of TLR-mediated pathogen detection, delaying the recruitment of immune cells and adaptive responses to the infection site [22]. In addition, PDIM has been suggested to be crucial for phagocytosis and in the rupture of phagosomes and mitochondrial membranes [22,23].

### 2.2. Protein Virulence Factors

Along with abundant lipids, the Mtb cell wall also contains proteins that directly affect the host’s immune response. Various reports have shown that protein families, such as PE, PPE, and lipoproteins, alter the host immune response. The effects include modulating cytokine production and arresting phagosome maturation, phagosome escape, autophagy, and cell death [2,16].

#### 2.2.1. Major Proteins Virulence Factors from Mtb

Mtb possesses an early secretory antigenic target (ESAT-6) secretion (ESX) system, also known as the type VII secretion system. Mtb has five secretion systems, or ESX, ranging from ESX-1 to ESX-5. The ESX-1, -3, and -5 are essential for mycobacterial virulence and regulate protein secretion and transport across the cytoplasmic membrane and complex mycobacterial cell wall [24].

ESX genes encode the ESX-type proteins EspA, EspB, EspC, and EspG, the secreted proteins ESAT-6 and CFP-10, PE-PPE family proteins, and the conserved components EccB, EccC, EccD, and MycP [25,26].

ESAT-6, delivered through the ESX system, is also known as EsxA or Rv3875. It is either secreted alone or in a complex with the chaperone CFP-10 (EsxB) [27]. It regulates Mtb’s colonization, survival, pathogenesis, and granuloma formation. Additionally, it has been implicated in inhibiting phagosome maturation and modulation of host cell death [28,29].

PE and PPE proteins are transported across the bacterial inner membranes by the ESX-1, ESX-3, and ESX-5 secretion systems through the PE and PPE domains [30]. The PE and PPE protein families have conserved Pro-Glu and Pro-Pro-Glu motifs in their N-terminal regions and are named after these conserved amino acid residues. It has been estimated that 10% of the Mtb genome encodes PE/PPE proteins, which participate in infection, antigenic variation, and host–pathogen interactions [2]. In addition, some PE-PPE members, such as PPE34, PE-PGRS11, PE-PGRS17, PE-PGRS33, PPE26, PPE57, and PPE60, interact with TLR2 to modulate the host innate immune response by inducing cytokine secretion, necrosis, and apoptosis, and enhancing mycobacterial survival [31,32].

#### 2.2.2. Major Lipoproteins Virulence Factors from Mtb

Lipoarabinomannan carrier protein (LprG) is a 27 kDa triacylated lipoprotein of the Mtb cell wall, which is involved in cell wall biosynthesis, and is essential for the expression of surface LAM [33]. LprG displays TLR2 agonist activity; it binds to LAM, PIMs, and LM, enhancing their recognition by TLRs. It also inhibits antigen processing in human macrophage MHC II [34] through TLR-2 signaling and inhibits and controls phagosome–lysosome fusion [35]. LpqH is a 19 kDa O-glycosylated and acylated glycolipoprotein and a primary cell wall antigen recognized by T cells. It induces T cell proliferation in vitro, stimulates DC maturation and autophagy, activates TLR-2 [36], and can affect the expression and antigen presentation of MHCII [37]. 

PstS1 is a 38-kDa glycolipoprotein that acts as a phosphate transporter recognized by the MR, promotes phagocytosis, reduces the production of reactive oxygen species (ROS), and induces tumor necrosis factor (TNF) and interleukin (IL)-6 production in human monocytes [38]. 

#### 2.2.3. Major Phosphatases and Kinases Virulence Factors from Mtb 

SapM and PtpA are phosphatases that participate in phagosomal arrest. SapM is a 28-kDa Mtb acid phosphatase found in the culture filtrate, indicating that it is secreted. SapM dephosphorylates phosphatidylinositol 3-phosphate (PI3P), an essential molecule for phagosomes to acquire lysosomal constituents; thus, its presence in the phagosome inhibits phagosome–lysosome fusion in macrophage cell lines from both murine and human origin (RAW 264.7 and THP-1, respectively) [39,40]. Moreover, SapM is essential for the pathogenesis of Mtb, and its inhibition leads to impaired mycobacterial growth in THP-1 cells and guinea pig tissues [40].

PtpA is a low-molecular-weight secreted protein tyrosine phosphatase (PTP) activator that dephosphorylates several proteins to suppress the immune response and, hence, is essential for Mtb survival and pathogenicity [41].

PtpA binds to ubiquitin (via a ubiquitin-interacting motif-like region) and dephosphorylates molecules, such as JNK, p38, and VPS33B. Specifically, dephosphorylated VPS33B excludes V-ATPase from phagosomes during Mtb infection and inhibits phagosome acidification [42,43]. Nuclear PtpA binds to host DNA and regulates the transcription of genes involved in host innate immunity, cell proliferation, and migration [42]. 

Mtb secretes two serine/threonine kinases with different effector functions, protein kinase E (PknE), which regulates Mtb survival response by inhibiting apoptosis [44], and protein kinase G (PknG), which modulates autophagy [45]. Both kinases aid in the intracellular survival of the pathogen, conferring the ability to suppress the host’s innate immune response. The serine/threonine kinase PknF is another bacterial transmembrane protein that inhibits the NLR family pyrin domain, containing three (NLRP3) inflammasomes [46].

Figure 1 shows the locations of some of the more critical virulence factors that alter cell death mechanisms in bacilli.

## 3. Cell Death Mechanisms Activated by Mtb Virulence Factors: The Good and the Bad

Cell death was initially classified into three main types based on morphological, biochemical, and immunological criteria: type I (apoptosis); type II (autophagy); and type III (necrosis) [47]. Currently, novel types of cell death are characterized by a more integrative profile, including trigger stimuli, molecular mechanisms, and morphological, enzymological, and immunological characteristics.

In 2018, the Nomenclature Committee on Cell Death (NCCD) proposed a new classification based on the mechanistic aspects of regulated cell death (RCD) and a definition based on the molecular and essential aspects. Some of these cell death types include intrinsic apoptosis, extrinsic apoptosis, mitochondrial permeability transition (MPT)-driven necrosis, necroptosis, ferroptosis, pyroptosis, autophagy-dependent cell death, parthanatos, endothelial cell death, NETotic cell death, lysosome-dependent cell death, immunogenic cell death, cellular senescence, and mitotic catastrophe [6]. 

Intriguingly, the molecular mechanisms involved in each RCD type have a considerable degree of interconnectivity, sharing key mediators. The novel signaling pathways involved in RCD are still being characterized.

Apoptosis and necrosis can be considered as light and shadows in the immune response. Apoptosis is a “double-edged sword“ for the host because Mtb can inhibit or activate apoptosis for its benefit. Grover et al. showed that apoptosis disseminates Mtb infection in lung granulomas during later stages [48]. Necrosis enhances bacterial replication and facilitates reinfection [49].

Recently, it was reported that Mtb virulence factors block molecules of the BCL-2 family to inhibit conventional apoptosis. Moreover, it also simultaneously alters the levels of molecules involved in inducing apoptosis (related to RE stress), necroptosis, and pyroptosis; thus, this phenomenon might possibly be a strategy for Mtb to survive [10].

### 3.1. Apoptosis 

Apoptosis is a highly regulated cell death process, wherein a broad definition of molecular mechanisms allows it to be classified as type I cell death. It is characterized by cytoplasmic contraction, confinement of cytoplasmic contents, chromatin condensation (pyknosis), nuclear fragmentation (karyorrhexis), and blebs in the plasma membrane, forming tiny, apparently intact vesicles (known as apoptotic bodies) that can be removed by phagocytosis [6]. Furthermore, the nature of the stimuli may trigger two different apoptotic pathways, the extrinsic and intrinsic (or mitochondrial) pathways, with convergence, overlap, and synergism in different steps, mainly in the effector caspases [6]. 

The factors initiating the intrinsic pathway include cytotoxic stresses, such as radiation, toxins, hypoxia, reactive oxygen species, virus infection, and cytokine deprivation [48]. Extracellular alterations induce the extrinsic pathway, where cell-surface death receptors, such as FAS (also CD95 or APO-1), members of the TNF receptor superfamily [6,50], and dependence receptors (such as netrin receptors DCC and UNC5B) trigger apoptosis in the absence of ligands [51].

Extrinsic apoptosis mediated by the oligomerization of death receptor signaling may occur in two ways: (a) caspase-8/-10 activation, leading to caspase-3 activation to cleave critical proteins for cell survival and DNA fragmentation [52]; and (b) caspase-8 activation leading to BID cleavage and mitochondrial outer membrane permeabilization (MOMP) induction. It has also been suggested that extrinsic apoptosis mediated by dependent receptors leads to the activation of caspase-9 and caspase-3 [51]. 

After ligand-death receptor interactions, monomeric caspase-8 is recruited through its death-effector domain (DED) to the death-inducing signal complex (DISC) in the cytoplasmic domain of the death receptor [53]. In addition, a conformational change allows the association of the adapter protein FAS-associated death domain (FADD) or TNF receptor-associated death domain (TRADD), leading to caspase-8 dimerization and activation of effector caspases, thereby impacting the methods previously described to induce cell death [6,53]. Caspase-8 autocleavage is essential for apoptosis; however, this process also inhibits necroptosis and affects lymphocyte homeostasis via the RIPK3/ mixed lineage kinase domain-like protein (MLKL) signaling axis [54].

In the intrinsic apoptosis, the BH3-only proteins activate the Bcl-2 family members BAX and BAK, leading to MOMP, impact on loss of mitochondrial transmembrane potential (ΔΨm), and induce release to cytosol apoptotic mitochondrial factors, such as cytochrome c, which interacts with Apaf-1 (Apoptotic protease activating factor-1) via their WD domain. This interaction induces conformational changes to induce APAF-1 oligomerization and interaction with dATP. Apaf-1-CARD domains are exposed in the apoptosome complex to recruit and activate initiator caspase-9. The assembly of CARD domains occurs in two kinds of complexes depending on apoptosome localization, which stimulates caspase-9 catalytic activity and suppresses inhibitory CARD activity in free caspase-9 [50,55]. Some pro-apoptotic proteins, such as SMAC and OMI, may be released by MOMP, inhibiting the caspase inhibitor XIAP (X-linked inhibitor of apoptosis protein). Generating tBID (a BH3-only protein) by caspase-8 may link the intrinsic and extrinsic apoptotic pathways. However, tBID may promote MOMP, and proapoptotic factors are released independently of BAX and BAK but positively impact SMAC [56,57].

#### 3.1.1. Modulation of Apoptosis in the Mtb Infection: Proteins Involved

Apoptosis is an RCD that helps eliminate the pathogen during mycobacterial infection. However, apoptosis can be a process with dual impact: on one side, it is a defense mechanism to control the pathogen, but on another side, it can be a strategy for its multiplication and dissemination [9].

Several virulence factors affecting apoptosis have been described in mycobacteria. For example, virulence factors of Mtb may lead to the capacity of infected cells to undergo or inhibit apoptosis. Macrophages may eliminate non-pathogenic mycobacteria, such as *M. fortuitum* and *M. smegmatis,* and induce an innate immune response [58]. In this regard, particular variations in mycobacterial strains influence their ability to affect the balance between cell death and proliferation [59].

It is well established that TNF signaling is essential to promote cell recruitment and granuloma formation. Inhibiting TNF by anti-TNF monoclonal antibodies, evidenced via in vitro models that change granuloma formation and structure, increases the risk of activating latent tuberculosis [60]. Furthermore, Mtb reactivation is dependent on TGF-β1 in particular in combination with Adalimumab, a fully humanized IgG1 TNF inhibitor [61]. 

Mtb-infected macrophages can up-regulate apoptotic signals through the differential expression of 42 genes belonging to the TNFR family [44]. All these pathways may be influenced by MAPSKs/NF-kB activity. Moreover, dual-specificity phosphatase 1 (DUSP1) has a relevant proapoptotic role in Bacillus Calmette–Guérin (BCG) infection [62]. Furthermore, the virulent strains of Mtb interrupt TNF death signals to affect apoptosis evolution, specifically, the DISC conformation or initiator caspase activation, even with low-dose infection [63]. It has been reported that soluble TNF Receptor-2 (TNFR2) blocks the axis TNF/TNFR1, reducing the Fas ligand expression [64].

TNF expression may be associated with the clearance of non-pathogenic strains in human alveolar macrophages and is induced during the primary immune response [65]. TNF also increases caspase-3, leading to a convergence of the intrinsic and extrinsic apoptotic pathways and the final consequences of apoptosis [66]. Soluble TNFR2 may inhibit induced apoptosis during the Mtb infection. However, this phenomenon was not observed during the BCG infection, suggesting that the impact of TNFR2 may be modulated by mycobacterial virulence [67,68]. In macrophages, TNF-induced ROS production increases the associated kinases, c-Abl, ASK1, and p38, leading to FLIP (FLICE-inhibitory protein) phosphorylation, promoting its interaction with the ubiquitin E3 ligase c-Cbl. FLIPS is ubiquitinated and degraded by the proteasome, allowing procaspase-8 and FADD interaction to trigger apoptosis [69].

The Mtb H37Rv strain was associated with the inhibition of apoptosis in U937 and THP-1 cells and human monocyte-derived macrophages, interfering with the extrinsic but not the intrinsic pathway. Furthermore, the putative proteins Rv3654c and Rv3655c have been implicated in inhibiting apoptosis via the extrinsic pathway [70]. Rv3654c interacts with PFS (polypyrimidine tract-binding Protein-associated Splicing Factor), a transcriptional regulator of caspase activity, whereas Rv3655c interacts with ALO17 associated with ALK (anaplastic lymphoma kinase). However, these proteins have been associated with anti-apoptotic activity [70]. The mitochondrial apoptosis pathway is not blocked; additionally, it has been suggested that it may induce necrosis in virulent mycobacteria when the cell death signal is from internally infected macrophages [71].

PE-PGRS33, the most extensively characterized PE family protein, is involved in different types of cell death; PE-PGRS33 co-localization in the host-cell mitochondria induces necrosis and apoptosis. PE_PGRS5 (Rv0297) leads to apoptosis through TLR-4-dependent endoplasmic reticulum stress via its PE domain [48,72]. 

In addition, ESAT-6, together with PDIMs, induced apoptosis [73]. ESAT-6 increases the gene expression of caspases 1, 3, 5, 7, and 8. Caspases -1 and -5 are associated with the inflammatory process, and the rest participate in the apoptotic pathways [74,75]. Moreover, ESAT-6 increases the miR-155 expression via the TLR2/NF-kB/ SOCS1 signaling axis for promoting apoptosis and impacts the proinflammatory response mediated by IL-6 and TNFs [76].

Among the anti-apoptotic Mtb genes, nuoG, a subunit of type I NADH dehydrogenase (type NADH-1), affects TNF apoptotic signaling by neutralizing NOX2-derived ROS. Thus, there is a close connection between NOX2, phagosomal ROS, and TNF-signaling in apoptosis in macrophages [77]. In a model to improve *Mycobacterium bovis* BCG vaccine efficacy, the nuoG gene was deleted, and the mutant was evaluated as an immunogen in C57BL/6 mice. After Mtb infection, the up-regulation of apoptosis, autophagy, and inflammasomes was identified, suggesting diverse roles for the antiapoptotic virulence gene nuoG in host–pathogen interactions that impact the induction of the immune response [78].

Mycobacterial factors that participate in redox balance, such as katG, sodA, PknE, and secA2, have been identified as anti-apoptotic agents [79]. The secA-2 system, an ATPase in the Sec secretion system, exports proteins required for bacterial cell growth [80]. Several studies have documented the antiapoptotic properties of secA2 [80,81].

Serine–threonine protein kinases (STPKs) are molecules involved in signal transduction, which play a role in Mtb survival. The serine/threonine kinase PknE has defined roles during NO exposure linked to infected cell apoptosis; PknE contributes to apoptosis resistance during nitrate stress [82]. In an infected macrophage model, the deletion of PknE induced high levels of apoptosis mediated by Bax, Bid, TP53, and caspases, suggesting the involvement of the intrinsic apoptotic pathway. Importantly, in this model, the authors identified the down-regulation of Mcl-1 and decreased TNF- and IL-6 levels [44]. Similarly, PstS1 (38 kDa lipoprotein) induces apoptosis via TNF and FasL [83]. This efficient signal affects the activation of intrinsic and extrinsic pathways, activating initiator procaspases -8, -9, and effector procaspase -3 [84].

#### 3.1.2. Modulation of Apoptosis in the Mtb Infection: Lipid Molecules Involved

Mycobacterial lipoglycans also play a significant role in the intrinsic apoptotic pathways [14,85]. Evidence suggests that TLR-2 senses LM/LAM, which affects apoptosis to varying degrees because non-pathogenic strains trigger apoptosis in contrast to pathogenic strains [86]. LM from Mtb and other Mycobacterium species, such as *M. kansasii* and *M. chelonae*, induce apoptosis in THP1 macrophages [86]. 

Regarding ManLAM, it has been shown that the expression of proapoptotic Bcl-2-associated X protein (Bax) is down-regulated in murine macrophages infected with *Mycobacterium bovis*, and the antiapoptotic B-cell CLL/lymphoma 2 (Bcl2) family member A1 (Bcl-2) expression is stabilized in *M. tuberculosis H37Ra* [85]. In addition, ManLAM inhibits apoptosis in one way because it can up-regulate Bcl-2; however, it also up-regulates FasL expression and down-regulates IL-12, which protects against apoptosis [85,87].

LpqH interacts with TLR2 and up-regulates death expression [88] in a dose-dependent model using LpqH, isolated from *M. smegmatis*, incubated with human macrophages isolated from peripheral blood from healthy individuals, induced loss of ΔΨm, the release of cytochrome c, and formation of apoptosome with the subsequent activation of procaspases-9 and -3 [89].

Figure 2 presents a general overview of apoptotic events and how Mtb virulence factors modulate them. 

In summary, the Mtb’s protein- and lipid-based molecules significantly affect apoptosis. Therefore, various efforts to activate or inhibit apoptosis have been implemented as therapeutic strategies to fight Mtb infections [90,91]. For example, in TB, the induction of apoptosis might be therapeutic, not only through the deregulation of the expression of virulence factors that inhibit the process but also through the direct activation of components, such as initiator or effector caspases, similar to other pathologies or health conditions [92]. 

### 3.2. Necrosis

Necrosis is considered to be a form of accidental cell death induced by pathological or physiological conditions, such as heat shock, mechanical stress, oxidative stress, inhibition of caspase activity, reduced levels of ATP, and radiation. This process encompasses several cell death modalities, commonly including the loss of plasma membrane integrity and the release of intracellular components that favor the inflammatory response, unlike that in apoptosis [7,93].

NCCD includes MPT-driven necrosis, a form of RCD induced by specific perturbations of the intracellular microenvironment triggered by severe oxidative stress and cytosolic Ca^2+^ overload [6]. An abrupt loss of the inner mitochondrial membrane’s impermeability to small solutes leads to a rapid Δψm dissipation and osmotic breakdown of both mitochondrial membranes and, consequently, RCD [94]. 

The presence of a necrotic morphology (i.e., rupture of the plasma membrane) can also be observed in the late stages of apoptotic or autophagic cell death by the absence of phagocytosis; apoptotic bodies may lose their integrity, and this is denominated as secondary necrosis caused by apoptosis or autophagy [95]. Secondary necrosis is not accidental but is controlled by a specific biochemical pathway regulated by caspase-3 and mediated by the GSDMD-related protein DFNA5 (deafness-associated tumor suppressor (GSDMD-related protein). Thus, this is a form of programmed necrosis similar to pyroptosis; caspase-3 cleaves DFNA5 to produce a necrosis-promoting DFNA5-N fragment that forms a large pore in the plasma membrane to release inflammatory molecules [96]. Other forms of regulated necrosis, including necroptosis, pyroptosis, and ferroptosis, have also been described.

#### Necrosis Modulation in Mycobacterial Infection

Macrophage necrosis is a strategy by which Mtb evades innate defense and could be a critical factor in forming granulomas, inflammatory tissue damage, and Mtb transmission [12]. Necrotic infected cells contribute to bacterial dissemination. Mtb inhibits plasma membrane repair, leading to necrosis and the release of mycobacteria into the extracellular environment [97,98]. Reports suggest that macrophage and neutrophil necrosis induced by Mtb contribute to its growth and sustained infection [99,100].

Among the virulence factors that activate necrosis, ESAT-6 induces neutrophil necrosis and NETs production by triggering intracellular Ca^2+^ influx [75]. The ESX-1 secretion system is essential in many virulence-related mechanisms to mediate macrophage necrosis, including host cell necrosis, extracellular trap formation, and mycobacterial aggregation [101]. PE17 (Rv1646) inhibits IL-6, IL-12, and TNF production, leading to macrophage necrosis [102]; as well, PPE11 (Rv0453) causes necrosis promoting mycobacterial survival [103]. Rv2626c interacts with PPE68 for secretion and is associated with host cell necrosis [104].

In a murine model, the accumulation of mycobacterial antigens in foamy macrophages from TB lesions was observed, which favors necrosis, tissue destruction, and mortality [105]. Figure 3 shows a diagram that summarizes how Mtb infection induces necrotic-like cell death, including pyroptosis, necroptosis, and ferroptosis, to promote disease progression [106,107].

### 3.3. Pyroptosis

Pyroptosis is a type of RCD that depends on the formation of pores in the plasma membrane by members of the gasdermin protein family and is often induced by the inflammatory activation of caspases [6]. Thus, this inflammatory form of programmed necrosis occurs mainly in myeloid cells after activating the pattern recognition receptor (PRR) [108]. In humans, it is triggered by caspase-1 and/or caspase-4/-5, inflammasome activation, and the maturation and release of I-β and IL-18 [109,110]. 

The inflammasome is a multiprotein complex containing PRRs, such as TLRs, nucleotide-binding domains, and leucine-rich repeat-containing receptors (NLRs), and is Absent in Melanoma (AIM)-like receptors (ALRs). It is activated by pathogen-associated molecular patterns (PAMPs) and danger-associated molecular patterns (DAMPs) when host cells are exposed to microbial infections, stress, or tissue damage [110]. NLRP1, NLRP3, AIM2, and pyrin require an adaptor protein for inflammasome assembly, an apoptosis-associated speck-like protein containing a caspase-recruitment domain (ASC), which facilitates the recruitment of pro-caspase-1 to the inflammasome complex [111].

NLRP3 is critical in bacterial, fungal, and viral infections and contributes to the host immune response [112]. The canonical activation of NRLP3 could be induced by two pathways: 1) by the detection of PAMPs, leading to up-regulated transcription of NLRP3, ASC, pro-IL-1β, pro-IL-18, and pro-caspase-11 by activation of NF-κB; or 2) by detection of DAMPS, such as abnormal levels of potassium (K^+^) efflux, chloride (Cl^−^) efflux, calcium (Ca2^+^) influx, oxidized mitochondrial DNA (ox-mtDNA), lysosomal rupture and intracellular ROS production. After recognizing PAMP or DAMP, the NLRP3 complex is oligomerized and assembled with ASC to recruit pro-caspase-1 for cleavage into active caspase-1. Finally, caspase-1 activates pro-IL-1β and pro-IL-18 via proteolytic cleavage into their active forms, IL-1β and IL-18, respectively [110]. 

As a result of inflammasome activation, caspase-1 cleaves Gasdermin D (GSDMD) into its pore-forming N-terminal fragment, GSDMD-N. Indeed, GSDMD is the primary executor of pyroptosis, forming pores in the plasma membrane resulting in pyroptosis [113,114]. Pyroptosis is another cell death mechanism that eliminates infected or compromised cells and plays a protective role in the host’s immune response [115].

#### Modulation of Pyroptosis by Mtb 

Mtb can manipulate pyroptosis and take advantage of it for survival, and mycobacteria modulate the host cell inflammasome [110]. 

NLRP3/AIM2 is activated by sensing Mtb effector molecules, such as ESX-1, LpqH, PPE13, EST12, and double-stranded RNA, as well as Mtb lipids, such as TDM and trehalose dimycolate (TDB) [110]. The role of the inflammasome is to limit Mtb growth; however, ESAT-6 has been reported to stimulate the activation of caspase-1 and the secretion of IL-1β to promote inflammasome activation and facilitate the spread of bacteria to neighboring cells [116,117,118]. Other Mtb proteins, such as PPE60 and PE_PGRS19, also induce pyroptosis by cleaving caspase-11, a non-classical pathway that probably benefits the spread of bacteria [119,120].

In contrast, Zmp1 and Rv3364c Mtb proteins inhibit pyroptosis. Zmp1 inhibits inflammasome activation by interacting with the mitochondrial respiratory chain complex 1 protein GRIM-19, an essential regulator of the NLRP3 inflammasome. Rv3364c translocates into the host cytosol to inhibit cathepsin G activity, affecting caspase-1 activation [121,122].

Recently, the phosphokinase PknF was implicated in inhibiting the NLRP3 inflammasome and inhibiting pyroptosis activation independently of the ESX-1 secretion system [46].

These findings suggest that Mtb may temporally regulate host cell death by inhibiting pyroptosis (and other cell death types) to establish an intracellular niche for replication. Once established, Mtb undergoes pyroptosis to escape and spread.

### 3.4. Necroptosis

Necroptosis is a highly regulated type of necrosis; this modality of RCD is triggered by perturbations of extracellular or intracellular homeostasis and is modulated by the kinase activity of the receptor-interacting protein 1 (RIPK1), RIPK3, and MLKL [6,123]. RIP Kinases play an essential role not only in inflammatory cell death even also in inflammatory signaling [124].

RIPK1 participates in TNF-induced NF-κB activation to regulate the switch between TNF-induced apoptosis and necroptosis, and when it is associated with RIPK3, necroptosis is induced [125,126].

Necroptosis can be triggered by impaired apoptosis, that is, the formation of necrosomes [127], by RIPK1-3; RIPK2 and RIPK3 are activated by TRADD, FAS, TNFR1, or PRRs. PPRs include TLR3, TLR4, and Z-DNA-binding protein 1 (ZBP1, also known as DAI) [125,128].

Additionally, RIPK3 can be activated following a domain-dependent interaction of the RIP-homotypic interaction motif (RHIM) due to the activation of TLR3 by double-stranded RNA (dsRNA) within endosomes and TLR4 activation by lipopolysaccharide (LPS) or DAMPs at the plasma membrane. RIPK3 catalyzes the phosphorylation of MLKL, leading to the formation of MLKL oligomers (trimers or tetramers) and their translocation to the plasma membrane, where they bind to specific phosphatidylinositol phosphate species [129,130]. 

MLKL promotes Ca^2+^ influx, probably by its target, the transient receptor potential cation channel, receptor subfamily M member 7 (TRPM7). When MLK is present on the membrane, the ADAM family proteases promote the shedding of membrane-associated proteins and form Mg^2+^ channels [131,132].

Some of the factors that regulate necrosome formation include stress-induced phosphoprotein 1 (STIP1) protein, homology, U-box containing protein 1 (STUB1 or CHIP), A20 protein, Mg^2+^/Mn^2+^-dependent protein phosphatase 1 B (PPM1B), aurora kinase A (AURKA), and RIPK3 [133,134].

STIP1 and STUB1/CHIP degrade RIPK1 and RIPK3 via ubiquitination and lysosomal degradation. A20 inhibits necrosome formation by eliminating RIPK3 ubiquitination. PPM1B prevents MLKL from binding to the necrosome by reducing RIPK3 phosphorylation. Finally, AURKA physically interacts with RIPK1 and RIPK3 to inhibit the necrosomal function [134,135].

These negative necrosome regulators play an essential role in regulating necroptosis and may modulate the pathogenesis of various diseases.

#### Modulation of Necroptosis by Mtb

Mtb employs virulence factors, such as the 38 kDa lipoprotein and ESAT-6, to induce TNF production, which induces necroptosis. High soluble levels of TNF and a high bacterial load indicate necroptosis instead of apoptosis [9,136].

Mtb triggers necrosis through tuberculosis necrotizing toxin (TNT), another secreted protein with NAD+-glycohydrolase activity that degrades all intracellular NAD+, activating necroptotic cell death via the RIPK3/MLKL pathway, independent of TNF signaling or RIPK1 [137,138]. These findings suggest that Mtb can induce different necroptotic pathways by switching the TNF/TNFR1/RIPK1 cascade to avoid immune responses.

ESX-1 also induces necroptosis. The ESAT-6 protein, together with the lipid PDIM, forms pores in the host membranes, causing destabilization and loss of integrity of the plasma membrane, which finally triggers necroptosis [139].

Mtb virulence also plays an essential role in the networking of cell death pathways displayed by the pathogen during infection. Mtb inhibits FADD expression from decreasing apoptosis, inducing necroptosis [140]. In addition, macrophages from patients with pulmonary TB infected with the Beijing strain developed severe pulmonary damage and inhibited FADD expression, exhibiting reduced apoptosis [141].

Afriyie-Asante et al. showed that host cells overexpress focal adhesion kinase (FAK) in response to mycobacterial infection. This cytoplasmic non-receptor protein tyrosine kinase promotes bacterial survival by inhibiting necroptosis and increasing ROS production [107]. 

Additionally, prevention of necrosome assembly and necroptosis are induced by caspase 8-mediated degradation of necrosome components, including the deubiquitinase CYLD, an essential component for necroptosis, which may be induced by cFLIP of cellular FLICE-like inhibitory protein (cFLIP). Although necroptosis is primed in Mtb-infected macrophages, necroptotic signaling is interrupted, inhibiting the cell death pathway [128].

The molecular mechanisms underlying necroptosis induced by mycobacterial infections remain poorly understood. More studies are necessary to elucidate how Mtb modulates necroptosis through its virulence factors and how the host immune response can abrogate it.

### 3.5. Autophagy-Dependent Cell Death

Autophagy-dependent cell death is a form of RCD that mechanistically depends on the autophagic machinery [6], an evolutionarily conserved catabolic process of eukaryotes, to maintain homeostasis in response to different stress conditions, such as starvation, hypoxia, absence of growth factors, and infection [142]. During autophagy, double-membrane vesicles, called autophagosomes, fuse with acidic compartments, or lysosomes, to give rise to autolysosomes that remove unwanted components from the cell, such as long half-life proteins, damaged organelles, and intracellular pathogens, such as Mtb. This phenomenon allows the cells to economize on macromolecule synthesis and increases cell survival [142,143].

Three types of autophagy have been described: chaperone-mediated autophagy (CMA); microautophagy; and macroautophagy. During CMA, cytosolic chaperones interact with soluble proteins and bind to the lysosomal receptor LAMP-2A (lysosome-associated membrane protein type 2a) for transportation to the lysosome and degradation of proteins [142,144]. Microautophagy involves the direct immersion of the material to be degraded by the lysosome through the lysosomal membrane’s extension, protrusion, and septation [145]. Macroautophagy, also known as the classical autophagy pathway, sequesters cytoplasmic portions and is named according to the content or organelle to be eliminated; for instance, when it contains intracellular pathogens, it is called xenophagy; when it contains mitochondria, it is called mitophagy and lysophagy [143,146,147]. 

#### Modulation of Autophagy by Mtb

Several Mtb virulence factors primarily prevent autophagy by arresting the fusion of phagosomes with lysosomes. For example, Eis, ESAT-6, PtpA, and PKnG are among the factors secreted by Mtb into phagosomes. These factors inhibit autophagosome formation [9,148].

PE6, another virulence factor, suppresses autophagy by inhibiting ULK1 or activating the phosphorylation of the autophagy master regulator MtorC1. This observation was reflected in the reduced conversion of the autophagy marker Microtubule-associated protein 1A/1B-light chain 3 (LC3) BI to LC3BII and the increased accumulation of the autophagy substrate p6T2. Furthermore, this suppression is also TLR4-dependent [149,150].

Mtb also inhibits the recruitment of Rab7, a critical component of mycobacteria-containing autophagosomes that mature into autolysosomes. Rab7 recruitment is notably affected by ESAT-6 and SapM proteins [145,151]. Moreover, the secreted SapM and PknG proteins enter the cytoplasm through EsxA-mediated phagosome disruption, further interfering with autophagosome maturation [152].

In addition, lipids from Mtb can also inhibit autophagy; for example, glycosylated PI, an analog of LAM, inhibits PI(3)P production by the Vps34 protein, whereas the SapM protein dephosphorylates PI(3)P to arrest phagosome maturation and increase Mtb survival [146,153].

Mtb uses several strategies to evade the host immune system and promote its survival. miR-889-5p expression inhibits autophagy and promotes bacterial survival. Similarly, miR-125a and miR-30a target UVRAG and Beclin-1, respectively, to inhibit autophagy [143].

Although different drugs, such as isoniazid, carbamazepine, loperamide, verapamil, valproic acid, and metformin, can activate the autophagy mechanism, the challenge remains to eliminate mycobacteria effectively. Further research is needed to understand the complexity of autophagy regulation in Mtb infection and to develop effective therapeutic interventions.

### 3.6. Ferroptosis

Ferroptosis is another form of RCD triggered by oxidative perturbations of the intracellular microenvironment under the constitutive control of glutathione peroxidase 4 (GPX4). Severe lipid peroxidation mainly depends on ROS generation and iron availability. Iron chelators and lipophilic antioxidants can inhibit ferroptosis [6].

Ferroptosis is regulated by molecules of the metabolic pathways that regulate cysteine exploitation, glutathione status, and nicotinamide adenine dinucleotide phosphate (NADP) function. Unlike other forms of RCD, such as apoptosis and necroptosis, ferroptosis is independent of caspases, necrosome components, and autophagy. Instead, ferroptotic cells display necrotic morphology with mitochondrial alterations, shrinkage, and electrodense ultrastructure. Additionally, they may release damage-associated molecular patterns (DAMPs) [154,155].

#### Modulation of Ferroptosis by Mtb

During Mtb infection, ferroptosis is triggered by various stimuli, including oxidative stress, lipid peroxide accumulation, and GPX4 inactivation or depletion [154,156]. Furthermore, ferroptosis contributes to tissue necrosis during TB infection.

The ferroptosis signaling pathway in Mtb infection involves several mechanisms. Heme oxygenase-1 (HO-1) regulates the expression of proteins, such as ferritin and ferroportin, and also mediates the immune response to TB infection. Nutritional immunity, which seeks to limit iron availability for invading microorganisms, plays an essential role in innate immunity [155,157].

The accumulation of ROS and lipid peroxides promotes tissue damage and bacterial dissemination, whereas HO-1 increases the levels of ferrous metals, contributing to Mtb survival [154]. Additionally, regulating ferritin and ferroportin by HO-1 can reduce intracellular oxidative stress and promote iron exit from the intracellular environment through the ferroportin protein [155,158]. Qiang et al. showed that PtpA triggers ferroptosis to promote Mtb pathogenicity and dissemination. PtpA inhibits GPX4 expression by inducing the formation of histone H3 arginine 2 (H3R2me2a), which, in turn, recruits methyltransferase 6 (PRMT6) to specifically down-regulate GPX4 expression, a key inhibitor of ferroptosis [159].

HO-1 and peptidyl-prolyl isomerase A (PPiA) are involved in Mtb survival. HO-1 can increase ferrous metal levels, whereas PPiA favors intracellular bacterial survival. Deleting PPiA in a murine infection model formed granuloma-like lesions and induced host cell death through ferroptosis [160].

Ferroptosis, which can be suppressed by inhibitors, such as ferrostatins and liproxstatins, is also involved in the oxidation of specific polyunsaturated fatty acids (PUFAs) containing phosphatidylethanolamines, such as arachidonic acid and adrenic acid. These oxidized PUFAs accumulate after GPX4 inactivation and may be involved in the inflammatory processes. Recent studies showed that specific inhibitors, including sorafenib, trigger ferroptosis [156,161,162,163,164].

The suppressor of cytokine signaling 1 (SOCS1) is the only gene related to ferroptosis in patients with TB and may regulate the microenvironment and bone destruction during TB infection. In addition, the release of damage-associated molecules and alarmins by the ferroptotic cells regulates immunity and pro-inflammatory activities during TB infection [165].

The PI3K signaling pathway may also regulate cell proliferation and growth by activating protein kinase B (Akt kinase, also known as PKB). Moreover, the p38 MAPK, ERK regulators, and JNK may participate in the intracellular signals that can induce HO-1 expression [155]. In contrast, the HO-1 enzyme has a dual role in TB, and its expression and activity can be therapeutically regulated. Finally, analysis of SOCS1_TB (profile GSE83456) identified 105 differentially expressed genes (DEGs), of which 97 were up-regulated, and six were down-regulated [165].

Therefore, ferroptosis plays an essential role in the pathogenesis of Mtb infection. Understanding the complex and multifaceted processes of ferroptosis regulation in TB is crucial for developing new therapies against TB and other infectious diseases. Identifying HO-1 and SOCS1 as potential therapeutic targets offers promising opportunities for the modulation of ferroptosis and for developing novel treatment strategies.

## 4. Conclusions

It is well-established that Mtb exploits its varied virulence factors to orchestrate the infection process, facilitating its growth, dissemination, and latency. Host cell death is a critical mechanism that determines the outcome of infection. It is one of the main processes manipulated by mycobacterium to favor itself, thereby having various consequences on the host cell. It has been proposed that apoptosis and pyroptosis, which are the cell death mechanisms helpful for Mtb, restrict intracellular bacterial growth and facilitate anti-Mtb immune responses. Meanwhile, necroptosis and ferroptosis benefit Mtb replication and transmission. 

As shown in Figure 4 and Table 1, the activation and inhibition of host cell death are driven by Mtb through its virulence factors; in fact, one virulence factor can be critical to more than one cell death modality, allowing the persistence of Mtb in the host. Among the virulence factors, ESAT-6 is essential in Mtb-modulated host cell death. It can trigger apoptosis, necrosis, necroptosis, and pyroptosis, mainly favoring the spread of mycobacteria. It can also inhibit phagosome maturation, thereby affecting autophagy, an essential mechanism for eliminating Mtb. Thus, ESAT-6 is one of the main tools used by Mtb to disseminate and survive in host cells.

A complete understanding of how Mtb interacts with its host to manipulate cell death mechanisms is still in progress. Therefore, new insights and findings will help understand why mycobacterium uses a specific mode of cell death and how it confers its virulence to escape the host immune response if simultaneously activated by diverse cell death mechanisms. 

## Figures and Tables

**Figure 1 pathogens-12-00839-f001:**
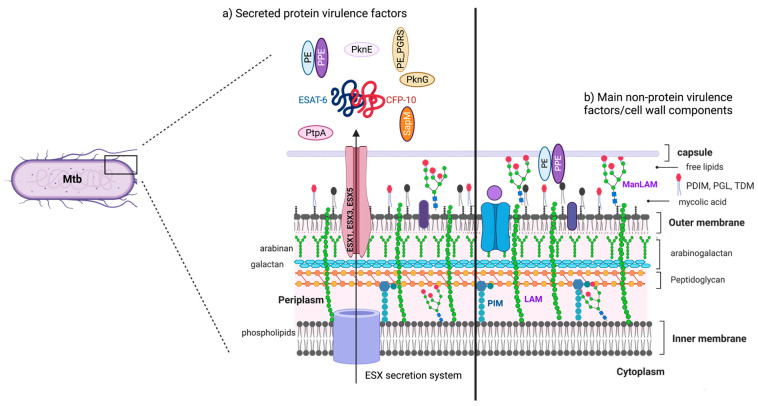
Major virulent factors of Mtb involved in cell death mechanisms. (**a**) Mycobacterial virulence proteins are mainly secreted through one of the essential secretion systems, ESX. ESX1, ESX3, and ESX5 are localized across the membrane, where proteins, such as ESAT-6, CFP-10, PE_PGRS, PtpA, SapM, PknG, and PknE, are delivered (left). (**b**) Non-proteins are glycolipids and phospholipids forming mycobacterial membrane structures. LAM (lipoarabinomannan), ManLAM (mannose-capped LAM), PDIM (phthiocerol dimycocerosate), and PIMs (Phosphatidyl-myo-inositol mannosides) are inserted in the cell wall (right). TDM, trehalose mono, and di mycolate; PGL, phenolic glycolipid.

**Figure 2 pathogens-12-00839-f002:**
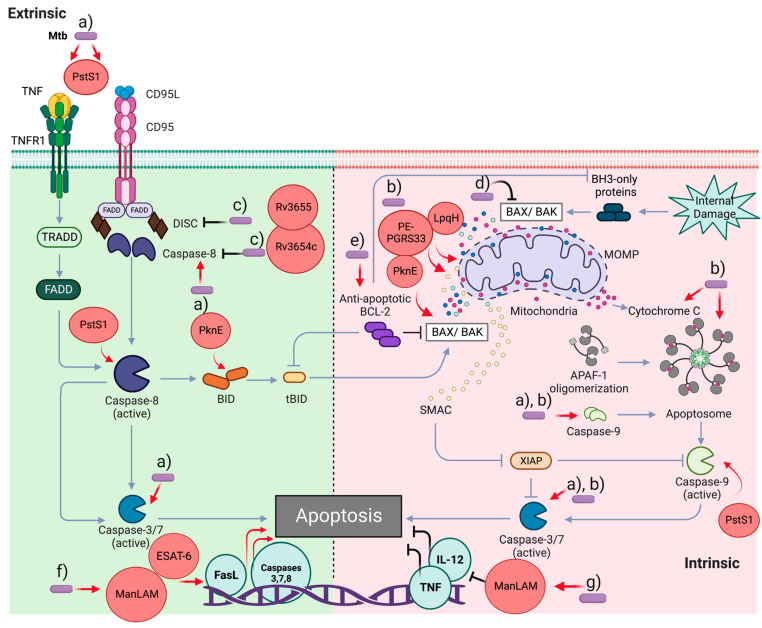
*Mycobacterium tuberculosis* and apoptosis modulation. The mycobacteria or its virulence factors may modulate apoptosis, both intrinsic and extrinsic pathways at different levels, as inductors (red arrows) or inhibitors (black inhibitors). (**a**) Apoptosis may be detonated via TNF and FasL impacting initiator caspases-8 and -9 and effector caspases-3 and -7. (**b**) The intrinsic pathway also may control mycobacteria infection at the mitochondria level, cytochrome c released, and apoptosome formation by activating caspases-9 and -3. (**c**) TNF death signal may be blocked at DISC formation and caspase-8 activation. (**d**) The proapoptotic protein BAX may be down-regulated with the H37Ra strain. (**e**) Mycobacteria may stabilize the anti-apoptotic Bcl-2. (**f**) Up-regulation at the gene level of FasL and caspases may be done during mycobacteria infection; (**g**) down-regulation of IL-12 expression and TNF has also been reported.

**Figure 3 pathogens-12-00839-f003:**
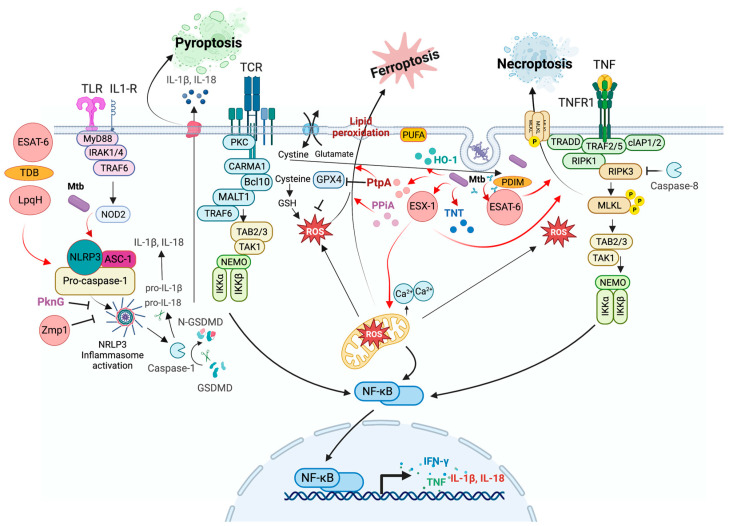
*Mycobacterium tuberculosis* and the necrotic-like cell death modulation. The mycobacteria or its virulence factors may modulate pyroptosis, necroptosis, and ferroptosis pathways at different levels as inducers (red arrows) or inhibitors (black inhibitors). For example, Mtb through ESAT-6, LpqH activates inflammasome, favoring pyroptosis, while Zmp1 and PknF inhibit it. The Mtb ESX-1 secretion system, secreted protein ESAT-6, TNT, and PDIM lipid induce necroptosis. Meanwhile, multiple factors inhibit or activate ferroptosis, including the HO-1 enzyme, the SOCS1 protein, and the PI3K/Akt/mTORC1 signaling pathway, which act at different levels and points regulating the cell death pathway. PtpA favors ferroptosis by inhibiting GPX4 expression, a ferroptosis regulator.

**Figure 4 pathogens-12-00839-f004:**
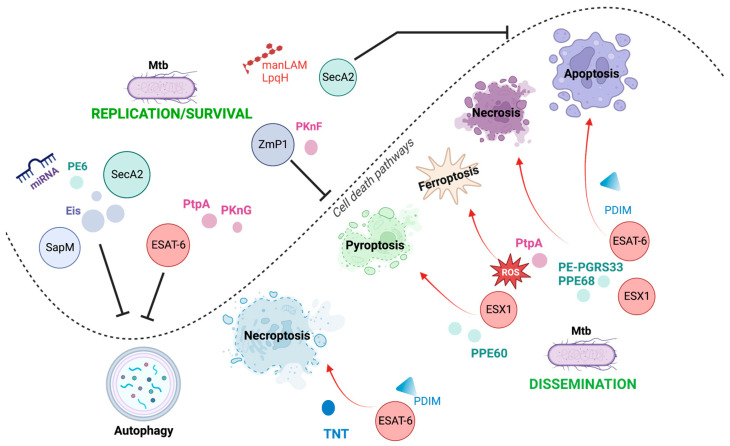
Overview of cell death pathways modulated by Mtb through their virulence factors. Mtb can use its various virulence factors as inductors (red arrows) or inhibitors (black inhibitors) of diverse cell death mechanisms to orchestrate the infection process; as well, it allows it to disseminate and replicate for its survival.

**Table 1 pathogens-12-00839-t001:** Mtb virulence factors involved in cell death modulation and its survival impact.

Mtb Virulence Factor/Protein or Lipid Origin	Abbreviation	Function/Impact on Cell Death	Cell Type	Dissemination/ Intracellular Survival	References
Dual-Specificity Phosphatase 1	DUSP1	Proapoptotic, a regulator of MAPKs/NF-κB signaling pathway, induces apoptosis	THP-1 macrophages	Undefined	[62]
Rv3654c and Rv3655c putative proteins		Antiapoptotic interfering with the extrinsic pathway	Human monocytic cell line U937	Undefined	[70]
PE family proteins	PE-PGRS33, PE_PGRS5 (Rv0297)	Proapoptotic and induces necrosis	RAW 264.7 murine macrophages	Undefined	[72]
PE family proteins	PE17 (Rv1646)	Inhibits IL-6, IL-12, and TNF production and induces macrophage necrosis.	Mouse peritoneal macrophages, THP-1 macrophages, J774A.1 mouse cells, and BMDM C57BL/6		
	PPE11 (Rv0453)	Induces macrophage necrosis		Intracellular survival	[102,103,104]
	PPE68-Rv2626c	Induces necrosis			[119,120]
	PPE60, and PE_PGRS19	Induces pyroptosis by a non-classical-pathway		Dissemination	[149,150]
	PE6	Suppresses Autophagy			
Early secretory antigenic target 6 kD	ESAT-6 (EsxA)	Proapoptotic, increases gene expression of caspases-3, -5, -7, and -8. Promotes inflammasome activation-pyroptosis induction. Induces neutrophil necrosis and NETs production, and loss of integrity of the plasma membrane and triggers necroptosis Inhibits phagosome maturation	hMDM and THP-1 macrophages		[73,74,75]
					[116,117,118]
				Dissemination	[28,29]
	Zmp1, and Rv3364c	Inhibit inflammasome activation and pyroptosis			[121,122]
Subunit of type I NADH dehydrogenase	nuoG	Antiapoptotic, inhibits apoptosis in the infected cells	THP-1 macrophages, BMDM BALB/c mice	Undefined	[77,78]
ATPase of the Sec secretion system	SecA2	Antiapoptotic, secretion of several proteins involved in blocking apoptosis		Undefined	[79]
Serine threonine protein kinases	PknE	Proapoptotic	THP-1 macrophages	Intracellular survival	[44,82]
	PknF	NLRP3 inflammasome inhibition	BMDM		[46]
Protein kinase G	PknG	Phagosome arresting, inhibition of phagosome–lysosome fusion	J774 macrophage cells	Intracellular survival	[45]
Phosphatase	SapM	Phagosome arresting, inhibition of phagosome–lysosome fusion	RAW 264.7 murine macrophages, THP-1 macrophages, and guinea pig tissues	Intracellular survival	[39,40]
Protein tyrosine phosphatase	PtpA	Phagosome arresting, inhibition phagosome acidification	THP-1 macrophages	Intracellular survival	[41,42,43]
		Triggers ferroptosis		Dissemination	[159]
38 kDa lipoprotein	PstS1	Proapoptotic, TLR-2 and caspases -8, -9, and -3 activation, and promotes phagocytosis	hMDM		[84]
Tuberculosis necrotizing toxin	TNT	Degrades all intracellular NAD+ and triggers necroptosis	THP-1 macrophages	Undefined	[137,138]
Enhanced intracellular survival protein	Eis	Abrogates ROS and proinflammatory cytokines production. Inhibits autophagosome formation	THP-1 macrophages	Undefined	[9,148]
Phthiocerol dimycocerosates	PDIMs	Proapoptotic, increases gene expression of caspases-3, -5, -7, and -8	hMDM	Undefined	[73]
		Induces necroptosis	THP-1 macrophages		[139]
Lipomannan (Glycolipid)	LM	Proapoptotic	THP-1 macrophages	Undefined	[86]
Mannosylated lipoarabinomannan (glycolipid)	ManLAM	Antiapoptotic, up-regulates Bcl-2 depending on the virulence	THP-1 macrophages, BMDM BALB/c mice	Undefined	[85,87]
Lipoprotein	LpqH	Proapoptotic, promotes apoptosome formation and activation of procaspases -9 and -3	hMDM	Undefined	[89]

hMDM, human monocyte-derived macrophages; BMDM, bone-marrow-derived macrophages; THP-1, human monocyte cell line.

## Data Availability

Not applicable.

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
