# Peer review of "Virulence Factors of Mycobacterium tuberculosis as Modulators of Cell Death Mechanisms"

_pathogens, 2023, doi:10.3390/pathogens12060839_

Round 1

Reviewer 1 Report

This review by Ramon-Luing et al is an appropriate compilation of virulence factors that affect cell death in mycobacterial infections. Overall, the authors have succinctly described all the virulence factors. However, the manuscript will benefit from the following changes: the most important one being to add a section for studies where information from virulence factors has been used to see host dissemination and survival.

Major comments:

1. Line 35-37: Reference for the fact that TB  death reductions were overestimated because of lack of diagnosis is missing. The reference (3) added here only talks about how COVID-19 affected essential TB services during this time.

2. Reference for line 46-47 is missing, for the part where the authors state that cell death is defensive process. 

3. It is recommended that the authors make a table for all the virulence factors and their respective mechanism of altering the cell death. The readers will also benefit from the understanding of what host system was used in a particular study - Mouse, guinea pig or human. Was it an animal study, organic study or cell culture study.

4. The readers will also beefier form a review section where researchers have used the information from all the other studies demonstrating mechanisms of alteration of cell death during Mtb infection to know if inhibition of certain pathways leads to decreased dissemination in the host or increased survival.

Minor comments:

1. MTB should be written as Mtb.

2. Line 145: LPrG should be LprG

3. the whole manuscript needs English editing.

the whole manuscript needs English editing especially for grammar. As an example:

The main virulence factors of MTB involved in altering cell death pathways are classified according to the origin; it is into non-protein (for instance, lipomannan) and protein 16 (such as the PE family and ESX secretion system). 

Author Response

Dear reviewer, thanks for your valuable observations; we appreciate your suggestions to improve our manuscript. 

  1. Line 35-37: Reference for the fact that TB...

R= Dear reviewer, we apologize for the misinterpretation. We corrected this part, and the same reference was included. We based on the last global report by the WHO in 2022, which highlights that the COVID-19 pandemic continues to have a detrimental impact on access to TB diagnosis and treatment and the burden of disease (lines 36-37).

  1. Reference for line 46-47 is missing, for the ...

R= The new refence #8 was included (Jorgensen, I., Rayamajhi, M. & Miao, E. https://doi.org/10.1038/nri.2016.147).

  1. It is recommended that the authors make a table for ...

R= We included table 1 in the new version, and a paragraph was included in the conclusion section (lines 657-666). 

  1. The readers will also beefier form a review section where ...

R= Dear reviewer, I don't know if I correctly interpreted your suggestion. We consider that your request is indicated throughout the manuscript, and the figures show the virulence factor and the inhibited pathway.

Minor comments:

  1. MTB should be written as Mtb.

The abbreviation was changed. 

  1. Line 145: LPrG should be LprG (line 138)

The change is done.

  1. the whole manuscript needs English editing.

The manuscript was revised by an expert company (Editage).

Reviewer 2 Report

1.      M. tuberculosis is abbreviated as MTB. To avoid confusion with TB, this should be modified as Mtb throughout the manuscript.

2.      Lines 64-67, the message that the authors want to convey is not clear. It is enough to say that M. tuberculosis doesn’t use toxins etc.

3.      Lines 69-70, please modify the sentence as “following are the Mtb molecules that can modify the cell death pathways”.

4.      Please avoid the word “Main” in 2.2.1, 2.2.2, 2.2.3 and under Figure 1. May use the word ‘major’ instead.

5.      Lines 181-182, please delete this sentence.

6.      Line 271, TB is misleading here. This may be Mtb.

7.      Section 3.1.1.  This section mixes the protein with lipids virulence factors in the manipulations of apoptosis. These may be separated for clarity. Please do this in other sections as well.

8.      Line 301. It is better not to mention about cancer here.

9.      Lines 455-1456. Move this sentence to the previous paragraph as the last sentence.

10.   Section 3.5.1, modulation of autophagy: This section should focus only on Mtb factors that interferes with autophagosomes formation. Phagolysosomal fusion is a common pathway for endosomes, phagosomes and autophagosomes.  Referring of Mtb factors that modulate phagolysosomal fusion here seems inappropriate. 

This manuscript needs to be edited by a native English speaker for clarity.

Author Response

Dear reviewer, thanks for your valuable observations; we appreciate your suggestions to improve our manuscript. 

  1. M. tuberculosis is abbreviated as MTB. To avoid confusion ...

R=Thank you. The new version has Mtb as an abbreviation.

  1. Lines 64-67, the message that the authors want to convey ...

R= To avoid this confusion, this expression was eliminated. 

  1. Lines 69-70, please modify the sentence as “following ...                            R= It was modified (lines 67-68).
  1. Please avoid the word “Main” in 2.2.1, 2.2.2, 2.2.3 and under Figure 1...

R= Main was replaced by major. 

  1. Lines 181-182, please delete this sentence.

R= Lines were deleted.

  1. Line 271, TB is misleading here. This may be Mtb.

R= It was rephrased. 

  1. Section 3.1.1.  This section mixes the protein with lipids virulence ...

R= Dear Reviewer, we appreciate this valuable suggestion. The apoptosis section was divided into 3.1.1 to describe the effect of proteins and 3.1.2 to describe the effect of the lipid. 

Here is important to note that apoptosis is the section that there are a lot of studies. Due to other cell death types having less described the effect of proteins and lipids, and the manuscript has described 1-2 small paragraphs, we consider that in these types of cell death is not necessary to divide into subsections.

  1. Line 301. It is better not to mention about cancer here.

R= The cancer sentence was deleted.

  1. Lines 455-1456. Move this sentence to the previous paragraph as the last sentence.

R= It is done (lines 443-444).

  1. Section 3.5.1, modulation of autophagy: This section should focus only on Mtb factors that interferes with autophagosomes formation. Phagolysosomal fusion is a common pathway for endosomes, phagosomes and autophagosomes.  Referring of Mtb factors that modulate phagolysosomal fusion here seems inappropriate. 

R= We appreciate this comment to improve our manuscript. This section was modified. 

Round 2

Reviewer 2 Report

I am satisfied with the revision.